# Counting Reward Automata: Sample Efficient Reinforcement Learning Through the Exploitation of Reward Function Structure

**Tristan Bester, Benjamin Rosman, Steven James, Geraud Nangue Tasse**

School of Computer Science and Applied Mathematics
University of the Witwatersrand
Johannesburg, South Africa
tristanbester@gmail.com, {benjamin.rosman1, steven.james}@wits.ac.za, geraudnt@gmail.com

## Abstract

We present counting reward automata—a finite state machine variant capable of modelling any reward function expressible as a formal language. Unlike previous approaches, which are limited to the expression of tasks as regular languages, our framework allows for tasks described by unrestricted grammars. We prove that an agent equipped with such an abstract machine is able to solve a larger set of tasks than those utilising current approaches. We show that this increase in expressive power does not come at the cost of increased automaton complexity. A selection of learning algorithms are presented which exploit automaton structure to improve sample efficiency. We show that the state machines required in our formulation can be specified from natural language task descriptions using large language models. Empirical results demonstrate that our method outperforms competing approaches in terms of sample efficiency, automaton complexity, and task completion.

## Introduction

In recent years, reinforcement learning (RL) has achieved a number of successes in complex domains ranging from video games (Badia et al. 2020) to robotics (Kumar, Todorov, and Levine 2016; Kalashnikov et al. 2018). However, these applications have largely been constrained to relatively simple, short-horizon tasks. This is a consequence of the techniques used for learning and reasoning, where approaches have been constrained to strategies relying on end-to-end learning systems. This restricts agents from accessing information about the structure of the problem which is known to designers (Marcus 2020). Consequently, the agent is required to unnecessarily learn information which could have been provided, resulting in poor sample efficiency (Marcus and Davis 2019; Arjona-Medina et al. 2019)

*Neuro-Symbolic Artificial Intelligence* is a subfield of AI that focuses on the integration of neural and symbolic methods with the intention of addressing the inherent weaknesses present in either of the approaches. It has been argued that the rich cognitive models necessary to solve temporally extended tasks require a combination of these approaches (Valiant 2008). Hierarchical reinforcement learning (HRL) and state machine-based approaches combine

symbolic reasoning with machine learning (Mitchener et al. 2022). These techniques are congruent with the "best of both worlds" perspective at the core of neuro-symbolic learning. Neural systems excel in training while remaining robust to noise in the data. Symbolic systems excel in the representation and manipulation of explicit knowledge (facilitating human-understandable explanations) while supporting provably correct reasoning procedures (Sarker et al. 2021).

While HRL (Barto and Mahadevan 2003) and state machine-based approaches (Icarte et al. 2018) have been proposed as potential solutions to learning long-horizon tasks, both methods are hindered by several shortcomings. HRL methods have traditionally been limited by practical challenges including exploration (Kulkarni et al. 2016) and reward definition (Eysenbach et al. 2018). In contrast, state machine-based approaches have shown significant potential in learning optimal policies for temporally extended tasks (Icarte et al. 2022). Unfortunately, these methods can only express a small set of tasks, limiting their application to a restricted set of problems.

We present a solution to long-horizon RL designed to address the limitations of current state-machine-based approaches. In this paper, we define a novel state machine variant known as a *Counting Reward Automaton (CRA)* capable of modelling any computer algorithm (Hopcroft and Ullman 1995). Our formulation is strongly reliant on both neural and symbolic methods. We rely on explicit symbol manipulation in the form of graph algorithms and natural language for task specification. We make the following contributions: (i) we propose counting reward automata as a novel abstract machine variant capable of modelling reward functions expressible in any formal language; (ii) we show that the state machines produced by this approach are both intuitive to specify through the use of *Large Language Models (LLMs)* as well as significantly simpler than those produced by alternative approaches; (iii) we describe how the sample efficiency of any off-policy algorithm can be improved through the use of counterfactual reasoning. A modified version of Q-learning is provided as an illustrative example; (iv) we discuss the conditions under which the proposed algorithms are guaranteed to converge to optimal policies and demonstrate their utility in several complex domains requiring long-horizon plans.

# Background

## MDPs, NDMPs, and RDPs

A *Markov Decision Process (MDP)* is a tuple $M = \langle S, A, T, \gamma, R \rangle$ where $S$ is the set of states which may be occupied by the agent, $A$ is the set of actions, $T : S \times A \to \Pi(S)$ is the transition function which returns a distribution over next states given action $a$ is executed in state $s$, $\gamma \in [0, 1]$ is a discount factor and $R : S \times A \to \mathbb{R}$ is a reward function representing the task to be solved. The agent is required to learn a Markov policy $\pi$ that performs action selection to maximise cumulative reward. The value function encodes the expected return associated with being in a given state. That is the expected total discounted reward when the agent is in a given state $s$ and selecting actions according to some policy $\pi$: $V^\pi(s) = \mathbb{E}^\pi[\sum_{t=0}^{\infty} \gamma^t R(s_t, a_t, s_{t+1})]$. An alternative value function is the action-value function $Q^\pi(s, a)$, which represents the expected total discounted reward when the agent executes action $a$ in state $s$ and follows policy $\pi$ thereafter.

A Regular Decision Process (Brafman, De Giacomo et al. 2019) is a restricted *Non-Markovian Decision Process (NMDP)*. An NDMP is defined identically to an MDP, except that the domains of $T$ and $R$ are finite sequences of states instead of single states: $T : S^+ \times A \times S \to \Pi(S)$ and $R : S^+ \times A \to \mathbb{R}$. In the RDP formulation, the dependence on history is restricted to regular functions.

## Reward Machines

A reward machine (RM) is a finite state machine that describes the internal structure of the reward function (Icarte et al. 2022). An RM operates by mapping abstracted descriptions of the current environment state to reward functions. This allows a single environmental interaction $\langle s, a, s' \rangle$ to be assigned different reward values based on the active RM state.

A reward machine makes use of a set of propositional symbols $\mathcal{P}$ which encode high-level events in an environment. The agent is able to detect the events in $\mathcal{P}$. The subset of events taking place in the environment at each time step is used as the input to the machine. The *labelling function* $L : S \times A \times S \mapsto 2^\mathcal{P}$ determines which events are taking place at a given instant in time. That is, the labelling function is used to assign truth values to the propositions in $\mathcal{P}$ given an environmental experience. A reward machine defined in an environment with states $S$ and actions $A$ is a tuple $\mathcal{R}_{\mathcal{PSA}} = \langle U, u_0, F, \delta_u, \delta_r \rangle$, where $U$ is a finite set of states, $u_0 \in U$ is the initial state, $F$ is a set of terminal states with $(U \cap F = \emptyset)$. The state-transition function is defined by the mapping $\delta_u : U \times 2^\mathcal{P} \mapsto U \cup F$. The state-reward function is given by the mapping $\delta_r : U \mapsto [S \times A \times S \mapsto \mathbb{R}]$. A reward machine $\mathcal{R}_{\mathcal{PSA}}$ begins in the initial state $u_0$. At each time step, the machine receives as input the set of high-level events taking place in the environment. Upon receiving this information, the machine transitions between states based on the state-transition function and outputs a reward function as defined by the state-reward function.

# Counting Reward Automata

To address the limited expressive power of reward machines, we introduce a novel abstract machine known as a *Counting Reward Automaton* (CRA) that is capable of modelling reward functions expressible as recursively enumerable languages. In contrast to RMs, this framework supports tasks described by unrestricted as opposed to regular grammars. Consequently, the formulation is compatible with any NMDP in which the dependence on history is expressible in a formal language. As a result, this framework may be applied to a much larger number of problems than RMs which are only applicable in RDPs.

## Counter Machines

Counter automata may be thought of as finite state automata augmented by a finite number of counter variables. While processing a string, the machine can update the values of its counters based on inputs, and the counters can in turn influence the machine's state transitions. A constraint imposed on the machine is that the value of its counters cannot be decremented below zero.

For $m \in \mathbb{Z}$, let $+m$ denote the function as defined in lambda calculus $\lambda x.x + m$. This function is used to modify counter values in response to machine transitions. A $k$-counter counter machine (Fischer, Meyer, and Rosenberg 1968) is defined as a tuple $\langle Q, F, \Sigma, \delta, q_0 \rangle$ where $Q$ is a finite set of machine states, $F$ is a finite set of terminal states, $\Sigma$ is the input alphabet, $q_0$ is the initial machine state and $\delta$ is the state transition function defined as

$$\delta : Q \times \{\Sigma \cup \{\varepsilon\}\} \times \{0, 1\}^k \to Q \times \{+m : m \in \mathbb{Z}\}^k$$

where $k$ is the number of counter variables and $\varepsilon$ is used to denote an empty string. The empty string allows the machine to transition without reading an input symbol.

A counter machine processes an input string one symbol at a time. For each input symbol, we use $\delta$ to update the machine configuration based on the input symbol and current machine configuration. A machine configuration is defined as $\langle q, \mathbf{c} \rangle \in Q \times \mathbb{N}^k$. Upon reading input symbol $\sigma$ the machine transitions according to

$$\langle q', \mathbf{c}' \rangle = \delta(q, \sigma, Z(\mathbf{c})),$$

where $Z : \mathbb{N}^k \mapsto \{0, 1\}^k$ is a *zero-test* function defined as

$$Z(\mathbf{c})_i := \begin{cases} 0 & \text{if } \mathbf{c}_i = 0 \\ 1 & \text{otherwise.} \end{cases}$$

A worked example showing how a counter machine operates has been included in the Appendix.

## Augmenting Agents with Counter Machines

We now introduce the CRA framework as an approach to modelling reward in NMDPs. We begin with the definition of a counting reward automaton. This is followed by the introduction of a running example, used to illustrate the operation of a CRA. Finally, we describe how the automaton can be thought of as defining a reward function on an MDP with a larger state space than that of the NDMP.

**Counting Reward Automaton** A CRA is a counter machine that has been augmented with an output function. The output function returns a reward function after each machine transition. A machine transition occurs every time the agent interacts with the environment. An abstract description of this interaction is used as the input to the machine. This causes the machine to transition, producing the reward function used to reward the agent for interaction.

**Definition 1.1** (Counting Reward Automaton)**.** *Given a set of environment states $S$, a set of actions $A$ and a set of propositional symbols modelling high-level events $\mathcal{P}$, a $k$-counter counting reward automaton is defined by a tuple $\langle U, F, \Sigma, \Delta, \delta, \lambda, u_0 \rangle$, where $U$ is a finite set of non-terminal states, $F$ is a finite set of terminal states ($U \cap F = \emptyset$), $\Sigma = 2^{\mathcal{P}}$ is the input alphabet, $\Delta = [S \times A \times S \mapsto \mathbb{R}]$ is the output alphabet, $\delta$ is the state transition function*

$$\delta : U \times \{\Sigma \cup \{\varepsilon\}\} \times \{0,1\}^k \mapsto \{U \cup F\} \times \{+m : m \in \mathbb{Z}^k\},$$

*$\lambda$ is the output function*

$$\lambda : U \times \{\Sigma \cup \{\varepsilon\}\} \times \{0,1\}^k \mapsto \Delta,$$

*and $u_0$ is the initial machine state.*

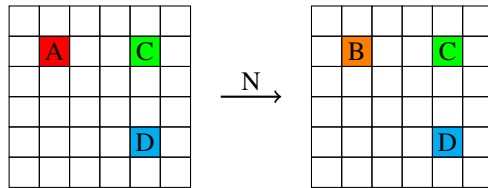

Figure 1: Illustration of the $LetterEnv$ environment, configured for the CFL experiment. The symbol A is replaced with a B after it has been observed $N$ times by the agent.

**Example Task** As a running example, we consider the *LetterEnv* environment presented in Figure 1. In this environment, a symbol (letter) is associated with certain positions. Symbols may either be observed infinitely often, or replaced by another symbol after a set number of observations. The agent is able to observe two aspects of the current state, its $xy$-location as well as the associated symbol (if such a symbol exists). The agent is able to move in any of the four cardinal directions.

For this example, the following environmental configuration is used. In each episode, the symbol $A$ may be observed a fixed number of times, before being replaced by the symbol $B$. The symbols $B, C$ and $D$ may be observed infinitely often. Specifically, the symbol $A$ may be observed $N$ times within an episode, where $N$ is a random variable with a discrete uniform distribution over the set $\mathbb{N}$. The agent is required to observe a sequence of environment symbols which corresponds to a string in the *context-free language (CFL)* described by the set $L = \{A^N B C^N : N \in \mathbb{N}\}$.

**Remark.** *As the reward model used in this example distinguishes between histories based on properties not expressible as regular expressions over the elements of the set $S \times A \times S$ (in this case counting how many times a state has been reached) it cannot be expressed as a reward machine.*

**CRA Operation** We use three propositions, one to model the presence of each symbol in the agent's current position $\mathcal{P} = \{P_A, P_B, P_C\}$. After each interaction of the agent with the environment $\langle s, a, s' \rangle$, the labelling function $L$ is used to compute the input to the machine $\sigma \in 2^{\mathcal{P}}$. Upon reading the input symbol $\sigma$, the machine transitions according to $\delta$ and emits a reward function based on $\lambda$. This process continues until the machine enters a terminal state.

We now describe how a CRA can be used to model the reward function for the example task. For convenience, we make use of a special case of the CRA formulation known as a *Constant Counting Reward Automaton (CCRA)*. After each transition, a CCRA returns a reward value directly as opposed to a reward function which is output by a CRA.

**Definition 1.2** (Constant Counting Reward Automaton)**.** *Given a set of propositional symbols modelling high-level events $\mathcal{P}$, a constant counting reward automaton is defined as a tuple $\langle U, F, \Sigma, \Delta, \delta, \lambda, u_0 \rangle$ where $U, F, \Sigma, \delta, \lambda$ and $u_0$ are defined as in a counting reward automaton; however, the output alphabet of the machine is defined as $\mathbb{R}$.*

**Theorem 1.** *For each constant counting reward automaton, there exists an equivalent counting reward automaton.*

*Proof.* See Appendix. □

A graphical representation of the example CCRA is illustrated in Figure 2. The machine contains two non-terminal states, one for each subtask in the specification. In the state $u_0$, the agent's objective is to observe the symbol $A$ repeatedly. However, in the state $u_1$, the agent is required to observe the symbol $C$ repeatedly. Intuitively, the automaton stores the number of $A$ symbols it has seen in its counter. After observing the symbol $B$, the machine decrements its counter each time the symbol $C$ is observed. Once the counter has been decremented to zero, the machine transitions into a terminal state and emits a reward of one. The automaton is represented as a directed graph. Each vertex in the graph represents a state in the machine, with $u_0$ being the initial state. Terminal states are represented by filled circles. Associated with each edge in the graph is a tuple $\langle \varphi, \boldsymbol{\omega}, \boldsymbol{\mu}, r \rangle$ in which $\varphi$ is a propositional logic formula over $\mathcal{P}$, the vector $\boldsymbol{\omega}$ contains the zero-tested counter states, $\boldsymbol{\mu}$ contains the counter modifiers and $r$ is the reward associated with the transition. A directed edge between the states $u_i$ and $u_j$ labelled by the tuple $\langle \varphi, \boldsymbol{\omega}, \boldsymbol{\mu}, r \rangle$ means that if the machine is in state $u_i$, the truth assignment $L(s, a, s') = \sigma$ satisfies $\varphi$, that is $\sigma \models \varphi$, and $Z(\boldsymbol{c}) = \boldsymbol{\omega}$, then the next machine state is equal to $u_j$ and the counter values are updated to $\boldsymbol{c} + \boldsymbol{\mu}$. For example, the edge between $u_0$ and $u_1$ labelled by $\langle P_B, [1], [0], 0 \rangle$ means that the machine will transition from state $u_0$ to state $u_1$ without modifying the counter value and emit a reward of one if the machine is in state $u_0$ with a non-zero counter value when the proposition $P_B$ becomes **true**. The automaton is updated after every agent-environment interaction. For example, if the agent takes action $a$ in state $s$ resulting in the following state $s'$, the machine configuration is updated to $\langle u', \boldsymbol{c}' \rangle = \delta(u, L(s, a, s'), Z(\boldsymbol{c}))$ and the agent receives a reward of $r = \lambda(u, L(s, a, s'), Z(\boldsymbol{c}))(s, a, s')$.

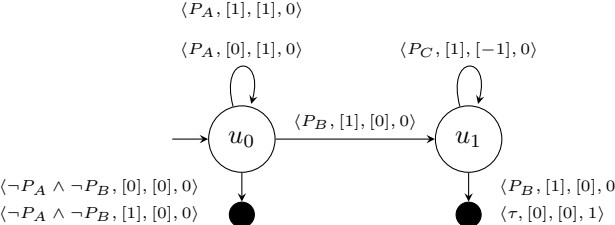

$$\langle P_A, [1], [1], 0 \rangle$$
$$\langle P_A, [0], [1], 0 \rangle \qquad \langle P_C, [1], [-1], 0 \rangle$$
$$\langle P_B, [1], [0], 0 \rangle$$

$$\langle \neg P_A \wedge \neg P_B, [0], [0], 0 \rangle \qquad \langle P_B, [1], [0], 0 \rangle$$
$$\langle \neg P_A \wedge \neg P_B, [1], [0], 0 \rangle \qquad \langle \tau, [0], [0], 1 \rangle$$

Figure 2: Illustration of a CCRA used to solve the example CFL task in the $LetterEnv$ environment. The $\tau$ symbol is used to represent a tautology (a propositional formula that is always true) which conditions the corresponding transition only on the states of the counters.

**Solving the Example Task**  With a CCRA in place, we proceed to discuss how a solution to the example task can be obtained. As described, the automaton specifies the reward function for the task. The reward is non-Markovian with respect to the ground environment state. That is, the agent's observation of the environment state does not contain sufficient information to determine the reward. For example, consider the case in which the agent observes that it is in the position associated with the symbol $B$. As this observation does not contain information about the sequence of symbols previously observed, it is insufficient to define the reward. However, the machine configuration can be combined with the ground environment state to produce a Markov state. As in the RM formulation, the automaton can be thought of as defining a reward function on an MDP with an alternate state space. Each state in this MDP is formed by combining an automaton configuration with a ground environment state. We refer to this MDP as the *Automaton-Augmented Markov Decision Process (AAMDP)*. While the general CRA formulation is not limited, we focus on finite-horizon MDPs. That is, we enforce a fixed upper bound on trajectory length. We use the symbol $\mathcal{H}$ to denote the maximum length of any trajectory. This value can be used to compute an upper bound on the value of any machine counter. Specifically, $\mathcal{H}$ is multiplied by the maximum counter increment defined in $\delta$ to produce the upper bound denoted $\Gamma$.

**Definition 1.3** (Automaton-Augmented Markov Decision Process). *An automaton-augmented Markov decision process is an MDP in which the reward function is modelled through the use of a counting reward automaton. An AAMDP is a tuple $\langle S, A, T, \gamma, U, F, \Sigma, \Delta, \delta, \lambda, u_0 \rangle$ in which $S, A, T$ and $\gamma$ are defined based on an environment representation, and $U, F, \Sigma, \Delta, \delta, \lambda$ and $u_0$ are defined as in a counting reward automaton. Each AAMDP induces an equivalent MDP $\langle S_{\mathcal{A}}, A_{\mathcal{A}}, T_{\mathcal{A}}, \gamma_{\mathcal{A}}, R_{\mathcal{A}} \rangle$ with $A_{\mathcal{A}} = A$, $\gamma_{\mathcal{A}} = \gamma$,*

$$S_{\mathcal{A}} = S \times \{U \cup F\} \times \{1, ..., \Gamma\}^k,$$
$$T_{\mathcal{A}} = P(\langle s', u', \boldsymbol{c'} \rangle \mid \langle s, u, \boldsymbol{c} \rangle, a)$$
$$= \begin{cases} P(s'|s, a) & \text{if } u \in F \text{ and } u' = u, \boldsymbol{c'} = \boldsymbol{c} \\ P(s'|s, a) & \text{if } u \notin F \text{ and } \langle u', \boldsymbol{c'} \rangle = \delta(u, \sigma, Z(\boldsymbol{c})) \\ 0 & \text{otherwise.} \end{cases}$$

*and $R_{\mathcal{A}}(s, a, s') = \lambda(u, \sigma, Z(\boldsymbol{c}))(s, a, s')$, where $\sigma := L(s, a, s')$.*

As the AAMDP is fundamentally an MDP, conventional reinforcement learning algorithms, such as Q-learning, may be used to compute a solution in the form of an optimal policy.

**Compatible Reward Functions**  As the CRA formulation is based on a counter machine, it is able to model both Markovian and non-Markovian reward functions to the extent that a Turing machine is able to distinguish between histories. This property holds as a two-counter counter automaton (equivalently, a two-stack pushdown automaton with a two-symbol alphabet) can simulate an arbitrary Turing machine (Hopcroft and Ullman 1995).

**Relationship between Counting Reward Automata and Reward Machines**  Reward machines are a special case of the CRA formulation. The *reward output function* of a reward machine is dependent only on the current machine state. As a result, an RM output function is equivalent to that of a CRA which returns the same reward function for all transitions out of a given machine state. The RM *state transition function* is dependent only on the current machine state and the input symbol. Consequently, the state transition function is equivalent to that of a CRA which does not modify the value of its counters.

**Theorem 2.** *Any reward machine can be emulated by a CRA with an equivalent number of machine states and transitions.*

*Proof.* See Appendix. $\square$

## Learning Algorithms

In this section, we discuss learning algorithms for the AAMDP formulation. As AAMDPs are fundamentally MDPs, we begin by discussing their compatibility with conventional reinforcement learning techniques. This is followed by the introduction of a novel learning algorithm for AAMDPs. This algorithm exploits task information encoded within the CRA of the AAMDP to increase sample efficiency. We provide pseudo-code implementations for the tabular case and discuss each algorithm's convergence guarantees.

### The AAMDP Baseline

Each AAMDP is an MDP which takes into account the state of the CRA at each time step. As a result, the AAMDP agent not only considers the underlying ground environment state $s$ when selecting an action, but also the CRA configuration $\langle u, \boldsymbol{c} \rangle$. Regardless of this distinction, the construction satisfies the properties of an MDP. Consequently, any learning algorithm designed to learn policies for an MDP is compatible with the AAMDP formulation. Any convergence guarantees associated with such algorithms in MDPs hold by extension in AAMDPs. As these algorithms do not use automaton information when learning, it is unlikely that any increase in sample efficiency will be observed when they are applied to AAMDPs. To illustrate how a conventional reinforcement learning algorithm can be used with an AAMDP, we provide an adapted version of tabular Q-learning in the Appendix.

## Counterfactual Experiences

We describe how task information encoded within the structure of a CRA can be used to enable more sample efficient learning. The learning algorithm we propose to exploit CRA task information is based on the *Counterfactual experiences for Reward Machines (CRM)* algorithm (Icarte et al. 2022). Our algorithm has been adapted to support the $k$-counter counter machine used in the CRA formulation. The counterfactual experiences generated by this algorithm can be used to increase the sample efficiency of any off-policy learning method.

We begin by providing an overview of the intuition behind the algorithm. We describe how task information, encoded within the automaton structure, can be used to enable more sample efficient learning. Suppose an agent in a ground environment state $s$ takes action $a$, producing a subsequent state $s'$. If the CRA configuration was $\langle u, c \rangle$ prior to the transition, then the reward signal emitted by automaton would be $r = \lambda(u, L(s, a, s'), Z(c))(s, a, s')$. However, we are able to consider every possible machine configuration prior to the transition. This counterfactual reasoning generates a set of counterfactual experiences, each of which may be used for learning. The generation of synthetic experiences is enabled by the task information encoded within the automaton structure.

In order to illustrate how counterfactual experiences can be incorporated into an off-policy learning algorithm, we present *Counterfactual Q-Learning (CQL)* in Algorithm 1. It can be seen that the only adaptation required is to the Q-function update. That is, rather than updating the Q-function once based on the observed AAMDP transition, it is updated multiple times, considering every possible prior automaton configuration. The process begins with the agent executing an action and observing the next ground environment state (line 8). This information is passed to the labelling function to compute the input to the machine. This is followed by counterfactual experience generation. That is, we simulate what would have happened had the machine been in any of its non-terminal configurations when the transition took place (lines 9-10). For each experience, the corresponding next machine configuration and reward are computed (lines 11-12). This information is used to update the Q-function (lines 13-17). Finally, the machine configuration is updated based on the transition that was observed in reality. We note that in practice, the loop on line 10 can be optimised by maintaining a cache of observed counter states.

**Theorem 3.** *Given an AAMDP $\mathcal{M}$, tabular CQL converges to an optimal policy for $\mathcal{M}$ in the limit (under the same conditions required for convergence of Q-learning in MDPs).*

*Proof.* As the counterfactual experiences generated in CQL are sampled according to the transition probability distribution $P(\langle s', u', c' \rangle \mid \langle s, u, c \rangle, a) = P(s'|s, a)$, the convergence proof provided by Watkins and Dayan (1992) for tabular Q-learning applies directly to the case of CQL. $\square$

---

**Algorithm 1: Counterfactual Q-Learning (CQL)**

**Input**: AAMDP $\mathcal{M} = \langle S_{\mathcal{A}}, A_{\mathcal{A}}, T_{\mathcal{A}}, \gamma_{\mathcal{A}}, R_{\mathcal{A}} \rangle$
**Output**: Optimal Q-value function $Q^*$

1: Initialise $Q(s, u, c, a) \leftarrow 0 \; \forall \; \langle s, u, c, a \rangle \in S_{\mathcal{A}} \times A_{\mathcal{A}}$
2: **repeat**
3:   Initialise $u \leftarrow u_0$
4:   Initialise $c \leftarrow \mathbf{0}$
5:   Initialise $s$ as initial environment state
6:   **while** $u \notin F$ and $s$ non-terminal **do**
7:     Sample $a$ from $\langle s, u, c \rangle$ using policy derived from $Q$ (e.g. $\epsilon$-greedy)
8:     Execute action $a$, observe $s'$
9:     **for** $u_i \in U$ **do**
10:       **for** $c_j \in \{1, ..., \Gamma\}^k$ **do**
11:         $\langle u_k, c_k \rangle \leftarrow \delta(u_i, L(s, a, s'), Z(c_j))$
12:         $r_k \leftarrow \lambda(u_i, L(s, a, s'), Z(c_j))$
13:         **if** $u_k \in F$ or $s'$ is terminal **then**
14:           $Q(s, u_i, c_j, a) \xleftarrow{\alpha} r_k$
15:         **else**
16:           $Q(s, u_i, c_j, a) \xleftarrow{\alpha} r_k + \gamma \max_{a' \in A} Q(s', u_k, c_k, a')$
17:         **end if**
18:       **end for**
19:     **end for**
20:     Compute $\langle u', c' \rangle \leftarrow \delta(u, L(s, a, s'), Z(c))$
21:     Set $u \leftarrow u'$
22:     Set $c \leftarrow c'$
23:   **end while**
24: **until** end

---

## Experimental Evaluation

In this section, we provide an empirical evaluation of the proposed methods. We begin by considering a *context-free* task specification that cannot be expressed by current state-machine-based approaches. In the following experiment, we compare the sample efficiency of our approach to that of state-of-the-art techniques for a *context-sensitive* task. Finally, we demonstrate how expert knowledge is naturally integrated into our formulation during task specification. An LLM is used to specify the desired task as a regular language which is used to produce the required automaton. A solution is obtained through function approximation.

### Task Specifications Beyond Regular Languages

We use the previously described *LetterEnv* environment to evaluate our methods on a task specification only expressible as a context-free language. The environment is illustrated in Figure 1. The agent is required to observe a sequence of environment symbols corresponding to a task in the context-free language $L = \{A^N BCD^N : N \in \mathbb{N}\}$. In our experiments, we compare CRA performance to that of state-of-the-art RM-based learning approaches (Icarte et al. 2022), including CRM and CRM with automated reward shaping (CRM+RS). For each approach, we compute the number of environmental interactions required to learn a policy which solves the task. The results are shown in Figure 3. As the

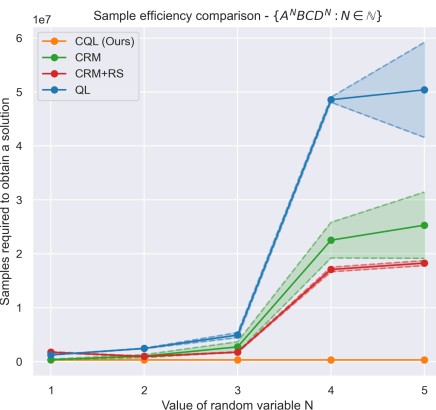

Figure 3: Number of samples required by each approach to obtain a solution to the task. Mean and variance are reported over 40 independent trials. A single CRA can be trained and immediately used to solve all tasks in the specification. For all other approaches, multiple policies must be learned.



Figure 4: Illustration of the *Office Gridworld* presented in Icarte et al. (2022). The agent, represented as a blue circle, begins in a fixed location. The agent is able to move in any of the four cardinal directions and its observations are restricted to its current position in the environment. The symbols ∗ represent decorations, which are broken if the agent collides with them. Mail can be collected from the location ✉ and coffee can be made at location ☕. A number of people are located at $P$. The trajectory for the context-sensitive task specification is shown

CRA is able to model the CFL reward directly, one machine can be trained and immediately used to solve all tasks in the specification. This is not true for the RM formulation, which is limited to the expression of reward functions in regular languages. This requires the RM formulation to emulate a CRA. This is achieved through the use of separate RMs for each value of $N$. Notably, while the CRA configuration remains constant across all tasks, the number of states required by an RM grows linearly with task complexity. The results demonstrate that the ability of the CRA to directly model more complex reward functions provides significant improvements in sample efficiency in comparison to CRA emulation through a collection of RMs.

## State Machine Complexity

We consider the *Office Gridworld* environment illustrated in Figure 4. We focus on a task specification that is only expressible as a context-sensitive language. The following task specification is used for this experiment. Firstly, the agent is required to navigate to the mail room and collect all available mail (the amount of mail is randomly generated at the start of each episode). The agent must then make a single coffee for each person that it has collected mail for. Finally, the mail and coffee must be delivered to the office employees located at $P$. The agent immediately fails the task if a decoration is broken or an incorrect number of coffees is collected/delivered. This task requires the agent to remember both the amount of mail it has collected as well as the number of coffees it has made. As a result of this property, this task can only be represented as a context-sensitive language. Due to the memory requirements associated with this task, two counter variables are used in the CRA formulation. For this task specification, we note that the CRA formulation, trained with the CQL algorithm, converges to a solution significantly faster than both CRM and CRM with automated reward shaping (CRM+RS) in all cases (see Appendix). However, we focus on the complexity of the state machines produced by either of the approaches. The CRA required to solve the task consists of three intuitive non-terminal states. These states correspond to the tasks: (i) collect all mail; (ii) make coffee and (iii) deliver the collected items. As the reward machine formulation is based on a finite state machine, it is unable to directly express task specifications corresponding to non-regular languages. This requires the reward machine to contain a single state for each CRA *task-counter* pair. We note that as the CRA formulation is able to express the context-sensitive task specification directly—it is therefore able to solve all task strings in

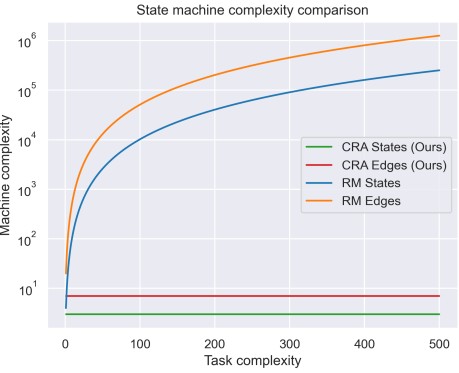

Figure 5: Comparison between the complexity of the state machines produced by the CRA and RM formulations. The illustration shows the complexity of the machine required to solve a task specification with a fixed upper bound on task-string length (in this case, the maximum number of mail items). RMs were constructed using a general template implementation parameterised by the maximum string length required for the task.

the language without modification. This is illustrated in Figure 5, with the machine configuration remaining constant as the maximum length of a task string is increased. An illustration of the state machines is provided in the Appendix. As existing approaches assume that the automaton is given, this requires that the specification of machines is intuitive and human-readable. We note that this is not the case. Complex algorithms were required to generate the RMs used in these experiments. The state machines required by current approaches are not only difficult to specify, but also to verify for correctness.

## Integration of Expert Knowledge in Learning

We now illustrate how expert knowledge can be naturally integrated into our formulation. We make use of the *Office Gridworld* environment to demonstrate our approach. We consider the following task specification. The agent is required to collect exactly one item of mail and deliver it to a person without breaking a decoration. The natural language description of this task is converted to a formal language through the use of an LLM, specifically ChatGPT (OpenAI 2023). The LLM is prompted to produce a formal language that specifies the sequence of high-level events satisfying the task description. This formal language description is used to produce the required automaton illustrated in Figure 7. The LLM prompts and responses are provided in the Appendix. Solutions to the task specification were obtained through function approximation, replacing Q-learning with *DQN* (Mnih et al. 2013). Full details of experimental hyperparameters a provided in the Appendix. The results are shown in Figure 6.

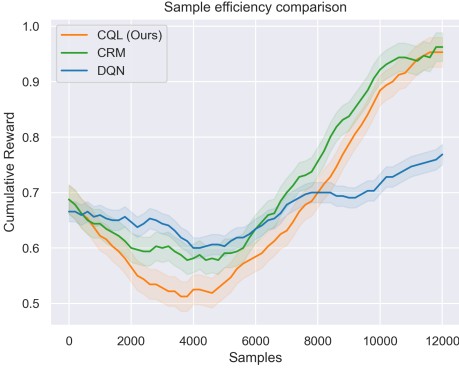

Figure 6: Sample efficiency comparison for natural language tasks solved through function approximation. Mean and variance were reported over 10 independent trials. CQL and CRM exhibit similar performance as the approaches are equivalent for regular language task descriptions.

## Related Work

In recent years, formal language and state machine-based approaches for task specification and efficient learning have been extensively studied (Littman et al. 2017; Li, Vasile, and

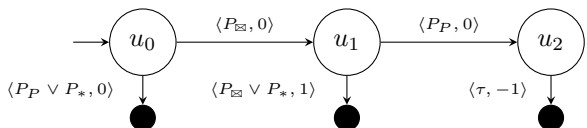

Figure 7: Illustration of a CCRA specified from a natural language task description through the use of an LLM. The propositions $P_s$ are satisfied when the symbol $s$ is encountered. As counter values remain unchanged for regular language tasks, they are not shown to save space.

Belta 2017; Brafman, De Giacomo, and Patrizi 2018; Jothimurugan, Alur, and Bastani 2019). This research has largely focused on the problems of specification like in Camacho et al. (2019) and learning the abstract machines from data like in Abadi and Brafman (2020). However, due to current formulations' dependence on *Finite State Machines (FSMs)*, they are significantly limited in terms of the types of reward functions they are able to express. This restricts the application of such methods to the small subset of problems which may be represented as *Regular Decision Processes (RDPs)* (Brafman, De Giacomo et al. 2019). This property imposes significant constraints on the classes of problems expressible by current approaches. For example, any problem which involves counting the number of times a state has been reached cannot be solved using these methods.

Finally, a related number of works have been proposed under the HRL framework for solving temporally extended tasks (Barto and Mahadevan 2003; Barreto et al. 2019). Solutions include methods such as the options framework (Sutton, Precup, and Singh 1999), HAMs (Parr and Russell 1997) and MAXQ (Dietterich 2000). Levy et al. (2017) make use of goal-conditioned policies at multiple layers of the hierarchy for RL. However, all of these approaches have been shown to struggle with exploration (Kulkarni et al. 2016) and reward definition (Eysenbach et al. 2018) which has limited their application.

## Conclusion

We have proposed the counting reward automata formulation as a universal approach to modelling reward in reinforcement learning systems. As a result of the formulation's dependence on counter machines, it is able to model reward functions expressible in any formal language, unlike previous approaches. Task information is encoded within the abstract machine during the reward modelling process, which may be exploited to enable more sample-efficient learning. Given a counting reward automaton for a task of interest, we have proposed a learning algorithm that can exploit the automaton structure to increase sample efficiency through counterfactual reasoning. The convergence guarantees of this approach have been discussed in the tabular case. Through empirical evaluation, we demonstrate the effectiveness of the proposed formulation both in terms of sample efficiency as well as state machine complexity. Finally, we have demonstrated how expert knowledge can be integrated into the formulation through the use of natural language task descriptions and LLMs.

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

# Appendix

## A. Counter Machine Example

In this section, we provide an example illustrating how counter machines operate. We describe how a counter machine can be used to recognise strings belonging to the context-free language $L = \{A^N B^N : N \in \mathbb{N}\}$. The counter machine is illustrated in Figure 8.

The machine is illustrated as a directed graph. Each vertex in the graph is a machine state and each edge in the graph is a transition. Associated with each edge is a label $\langle \sigma, \boldsymbol{\omega}, \boldsymbol{\mu} \rangle$ where $\sigma \in \Sigma$ is an input symbol, $\boldsymbol{\omega} \in \{0, 1\}$ is the *zero-tested* counter state and $\boldsymbol{\mu} \in \{+m : m \in \mathbb{Z}\}$ is the counter modifier.

We will use an example string $AABB$ to demonstrate how the machine functions. The machine begins in the initial state $u_0$ with a counter value of 0. The input string is processed one symbol at a time. Firstly, the symbol $A$ is read. At this point, the machine's counter state is $[0]$ as the counter value is zero. Upon reading this symbol, the machine transitions, remaining in state $u_0$ and incrementing its counter value by 1. Next, the symbol $A$ is read by the machine. At this point, the counter state is $[1]$ as the counter value is not equal to zero. This causes the machine to transition, remaining in state $u_0$ and incrementing its counter value to 2. The machine then reads the input symbol $B$. The machine transitions to state $u_1$ and decrements the counter value by 1. Finally, the machine reads the input symbol $B$. The machine transitions and decrements the counter value to zero. The input string is accepted as the automaton is in an accepting state with a counter value of zero after processing is complete.

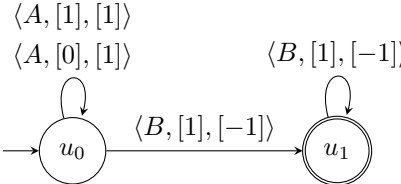

Figure 8: Illustration of a counter machine that recognises the language $L = \{A^N B^N : N \in \mathbb{N}\}$.

## B. Theorem 1.1

*Proof.* Let $\mathcal{A}_{\mathcal{P}} = \langle U, F, \Sigma, \Delta_c, \delta, \lambda_c, u_0 \rangle$ be a CCRA.

The equivalent CRA is defined as $\mathcal{A}_{\mathcal{PSA}} = \langle U, F, \Sigma, \Delta_f, \delta, \lambda_f, u_0 \rangle$ where $U, F, \Sigma, \delta$ and $u_0$ are defined as in the CCRA.

The output alphabet of the machine is defined as $\Delta_f = [S \times A \times S \rightarrow \mathbb{R}]$.

We now define the output function $\lambda_f$. A machine transition is defined for each triple $\langle u, \sigma, \boldsymbol{\omega} \rangle \in U \times \{\Sigma \cup \{\varepsilon\}\} \times \{0, 1\}^k$. Upon reading an input symbol, the machine transitions producing $\lambda_c(u, \sigma, \boldsymbol{\omega}) = x$ for some $x \in \mathbb{R}$. To emulate $\lambda_c$, the reward function returned by $\lambda_f$ for the transition is required to return $x$ for all inputs in the set $S \times A \times S$. As a result, $\lambda_f$ may return the function defined as

$$f_x : S \times A \times S \rightarrow \{x\}$$

$$f_x(s, a, s') = x \ \forall \ \langle s, a, s' \rangle \in S \times A \times S$$

In general,

$$\lambda_f(u, \sigma, \boldsymbol{\omega}) = f : S \times A \times S \rightarrow \{\lambda_c(u, \sigma, \boldsymbol{\omega})\}$$

$\square$

## C. Theorem 1.2

*Proof.* Let $\mathcal{R}_{\mathcal{PSA}} = \langle U^{\mathcal{R}}, u_0^{\mathcal{R}}, F^{\mathcal{R}}, \delta_u, \delta_r \rangle$ be a reward machine.

The number of RM states is equal to $|U^{\mathcal{R}}|$. By definition, an RM transition exists for each pair $\langle u, \sigma \rangle \in U^{\mathcal{R}} \times 2^{\mathcal{P}}$. It follows that the number of RM transitions is equal to $|U^{\mathcal{R}} \times 2^{\mathcal{P}}|$.

We now show how to define a CRA capable of emulating the RM. Firstly, to emulate the RM, the CRA does not require the use of its counters. Therefore, the counter values remain $\mathbf{0}$ at all times, as they are not modified in any transitions. Secondly, the CRA only transitions when the RM transitions, that is, after reading an input symbol. The CRA is defined as $\mathcal{A}_{\mathcal{PSA}} = \langle U^{\mathcal{A}}, F^{\mathcal{A}}, \Sigma, \Delta, \delta, \lambda, u_0^{\mathcal{A}} \rangle$ where $U^{\mathcal{A}} = U^{\mathcal{R}}$, $F^{\mathcal{A}} = F^{\mathcal{R}}$, $u_0^{\mathcal{A}} = u_0^{\mathcal{R}}$, $\Sigma = 2^{\mathcal{P}}$, $\Delta = Im(\delta_r)$,

$$\delta(u, \sigma, \mathbf{0}) = \langle \delta_u(u, \sigma), \mathbf{0} \rangle \ \ \forall \langle u, \sigma \rangle \in U^{\mathcal{A}} \times 2^{\mathcal{P}}$$

and

$$\lambda(u, \sigma, \mathbf{0}) = \delta_r(u) \ \ \forall \langle u, \sigma \rangle \in U^{\mathcal{A}} \times 2^{\mathcal{P}}$$

From $U^{\mathcal{A}} = U^{\mathcal{R}}$, it follows that the machines have the same number of states. As the domain of the CRA transition function $\delta$ is equal to $U^{\mathcal{A}} \times 2^{\mathcal{P}} = U^{\mathcal{R}} \times 2^{\mathcal{P}}$, it follows that the machines have the same number of transitions. $\square$

## D. Pseudo-code Implementation

---
Algorithm 2: Tabular Q-Learning for AAMDP
---
**Input**: AAMDP $\mathcal{M} = \langle S_{\mathcal{A}}, A_{\mathcal{A}}, T_{\mathcal{A}}, \gamma_{\mathcal{A}}, R_{\mathcal{A}} \rangle$
**Output**: Optimal Q-value function $Q^*$

1: Initialise $Q(s, u, \boldsymbol{c}, a) \leftarrow 0 \ \forall \langle s, u, \boldsymbol{c}, a \rangle \in S_{\mathcal{A}} \times A_{\mathcal{A}}$
2: **repeat**
3:     Initialise $u \leftarrow u_0$
4:     Initialise $\boldsymbol{c} \leftarrow \mathbf{0}$
5:     Initialise $s$ as initial environment state
6:     **while** $u \notin F$ and $s$ non-terminal **do**
7:         Sample $a$ from $\langle s, u, \boldsymbol{c} \rangle$ using policy derived from $Q$ (e.g. $\epsilon$-greedy)
8:         Execute action $a$, observe $s'$
9:         Compute $\langle u', \boldsymbol{c'} \rangle \leftarrow \delta(u, L(s, a, s'), Z(\boldsymbol{c}))$
10:        Compute $r = \lambda(u, L(s, a, s'), Z(\boldsymbol{c}))(s, a, s')$
11:        **if** $u' \in F$ or $s'$ is terminal **then**
12:           $Q(s, u, \boldsymbol{c}, a) \overset{\alpha}{\leftarrow} r$
13:        **else**
14:           $Q(s, u, \boldsymbol{c}, a) \overset{\alpha}{\leftarrow} r + \gamma \max_{a' \in A} Q(s', u', \boldsymbol{c'}, a')$
15:        **end if**
16:        Set $u \leftarrow u'$
17:        Set $\boldsymbol{c} \leftarrow \boldsymbol{c'}$
18:     **end while**
19: **until** end.
---

## E. Learning Curves for Office Gridworld

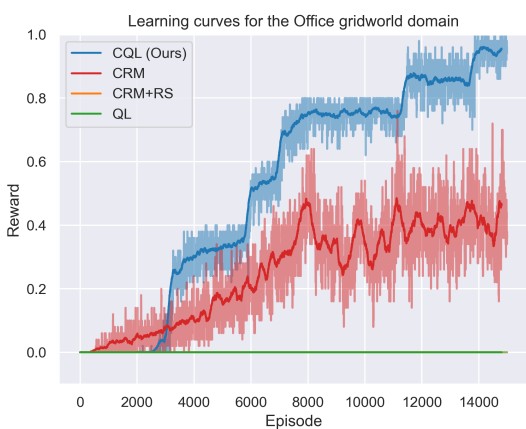

Figure 9: Learning curves for *Office Gridworld*. Mean and variance reported over 5 independent trials.

## F. Visualisation of Office Gridworld state machines

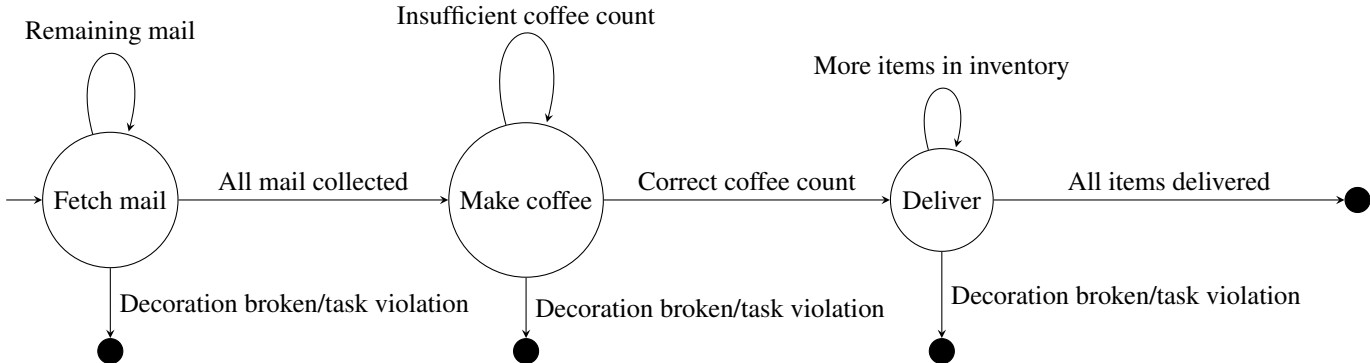

Figure 10: Illustration of simplified CCRA used to solve the *Office Gridworld* task for **any** value of $N$.

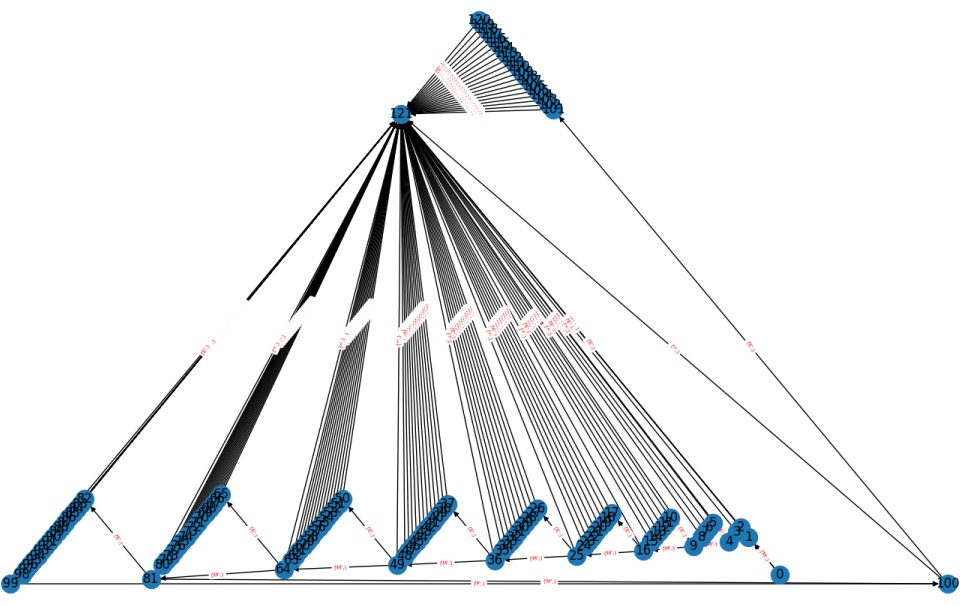

Figure 11: Illustration of RM used to solve the *Office Gridworld* task for $N = 10$.

## G. LLM Conversation
### 1. Formal Language Specification

**User:** The following events of interest have been defined: "M" mail is collected; "P" an item has been delivered to the person; "D" a decoration is broken.

A sequence of events can be represented as a string from a formal language.

Provide the formal language corresponding to the sequence of events: Collect exactly one item of mail and deliver a single item of mail to a person. Do not break a decoration at any point in the sequence.

**Bot:** Let's denote the events as follows:
- "M" for collecting mail
- "P" for delivering mail to a person
- "D" for breaking a decoration

The formal language corresponding to the given sequence of events can be represented as:

$$L = \{(MP)^1\}$$

This language specifies a sequence of events where exactly one "M" (mail collection) is followed by exactly one "P" (mail delivery to a person). The superscript 1 indicates that this sequence occurs exactly once, and there should be no "D" (decoration breaking) event in the sequence.

### 2. Automaton Specification

**User:** Describe an automaton that is able to recognise this language. This automaton should accept any strings if they are in the language and reject any strings if they are not in the language. Describe the transition function for the machine in each state for each input symbol in the alphabet.

**Bot:** To recognize the language $L = \{(MP)^1\}$, where exactly one "M" is followed by exactly one "P," without any "D" in between, you can design a simple deterministic finite automaton (DFA). The DFA will have three states corresponding to the different phases of the sequence: initial state, mail collected state, and mail delivered state.

The alphabet for this language consists of three symbols: $\Sigma = \{M, P, D\}$.

Here's the description of the automaton:

1. **States:**
   - $q_0$: Initial state
   - $q_1$: Mail collected state
   - $q_2$: Mail delivered state (Accepting state)

2. **Transition function:**
   - Transition from the initial state $q_0$:
     - On input $M$, move to state $q_1$ (mail collected).
     - On input $P$ or $D$, go to a trap state $q_{\text{trap}}$ (a non-accepting state indicating an invalid sequence).
   - Transition from the mail collected state $q_1$:
     - On input $P$, move to state $q_2$ (mail delivered, accepting state).
     - On input $M$ or $D$, go to $q_{\text{trap}}$.
   - Transition from the mail delivered state $q_2$:
     - On any input symbol $(M, P, D)$, go to $q_{\text{trap}}$ since the sequence is complete.
   - Transition from the trap state $q_{\text{trap}}$:
     - On any input symbol $(M, P, D)$, remain in $q_{\text{trap}}$.

3. **Accepting states:**
   - The only accepting state is $q_2$ (mail delivered state).

This DFA ensures that exactly one "M" is followed by exactly one "P," and any deviation results in the automaton entering a non-accepting trap state.

Here's a formal representation of the transition function:

$$\delta(q_0, M) \rightarrow q_1$$
$$\delta(q_0, P) \rightarrow q_{\text{trap}}$$
$$\delta(q_0, D) \rightarrow q_{\text{trap}}$$
$$\delta(q_1, M) \rightarrow q_{\text{trap}}$$
$$\delta(q_1, P) \rightarrow q_2$$
$$\delta(q_1, D) \rightarrow q_{\text{trap}}$$
$$\delta(q_2, M) \rightarrow q_{\text{trap}}$$
$$\delta(q_2, P) \rightarrow q_{\text{trap}}$$
$$\delta(q_2, D) \rightarrow q_{\text{trap}}$$
$$\delta(q_{\text{trap}}, M) \rightarrow q_{\text{trap}}$$
$$\delta(q_{\text{trap}}, P) \rightarrow q_{\text{trap}}$$
$$\delta(q_{\text{trap}}, D) \rightarrow q_{\text{trap}}$$

> Note: In the transition function, $\delta(q, a) \rightarrow q'$ denotes that on input symbol $a$ in state $q$, the automaton transitions to state $q'$.

## H. Function Approximation Hyperparamters

- We use feed-forward networks with 2 hidden layers.
- Hidden layers contain 128 neurons and make use of the rectified linear unit (ReLU) as an activation function.
- Networks are trained for 12,000 environmental interactions with a learning rate of $\eta = 10^{-4}$.
- On every optimisation step the Q-functions are using $32N$ sampled experiences from the replay buffer of size 50,000 where $N = 1$ for DQN and $N = |U|$ for CRM and CQL.