# OpenReview forum: "Counting Reward Automata: Sample Efficient Reinforcement Learning Through The Exploitation of Reward Function Structure"
_AAAI.org/2024/Workshop/NuCLeaR — NuCLeaR 2024_

### Official Review · Reviewer_YiPM · 2023-12-07
**Interesting paper, but less detailed information**

**Rating:** 6
**Confidence:** 3

**Review:**

Pros:
1. This paper presents an agent equipped with such an abstract machine is able to solve a larger set of tasks than those utilizing current approaches. Unlike previous approaches, which are limited to the expression of tasks as regular languages.
2. This paper shows that the state machines required in their formulation can be specified from natural language task descriptions using large language models. Empirical results demonstrate that their method outperforms competing approaches in terms of sample efficiency, automaton complexity, and task completion.

Cons:
1. The version of ChatGPT been used for evaluating in the paper did not been clearly mentioned. The experiment results in terms of sample efficiency from Figure 6 did not show explicitly better than the other baseline methods.
2. The experiment details include the description for the dataset such as train/dev/test sets, the name of the dataset, a detailed description or example for counterfactual reasoning are missing. Figure 6 says the mean and variance were reported over 10 independent trials. A confidence interval will be needed in this case.

---

### Official Review · Reviewer_LA7M · 2023-12-08
**Interesting ideas, less informative empirical evidence**

**Rating:** 6
**Confidence:** 3

**Review:**

### **Summary**
The paper presents counting reward automata (CRA), a novel framework for modeling reward functions in reinforcement learning (RL) as formal languages. The paper shows that CRA can express any recursively enumerable reward function, unlike previous state machine-based approaches limited to regular languages. The paper also proposes Counterfactual Q Learning that exploits the structure of CRA to improve sample efficiency and convergence guarantees. It demonstrates their effectiveness on complex tasks that require long-horizon planning and counting.
The paper’s strengths are:
It introduces a general and expressive framework for reward function specification that can handle a wide range of problems, including context-free and context-sensitive languages.
It provides theoretical and empirical evidence that CRA can improve the performance and scalability of RL agents compared to existing methods.

### **Strengths**
- General and expressive framework for reward function specification that can handle a wide range of problems, including context-free and context-sensitive languages.
- Comprehensive theoretical and empirical evidence that CRA can improve the performance and scalability of RL agents compared to existing methods.
- Leveraging natural language and large language models to facilitate the intuitive and human-readable specification of CRA from task descriptions.

### **Weakness**
- Lacking clear comparison or discussion of the trade-offs between CRA and other neuro-symbolic or hierarchical RL methods that also aim to address long-horizon tasks.
- Missing analysis or evaluation of the robustness, generalization, or interpretability of CRA-based agents.
- Somewhat toy examples

---

### Official Review · Reviewer_53Dk · 2023-12-08

**Rating:** 7
**Confidence:** 3

**Review:**

This paper introduces a novel finite state machine variant called Counting Reward Automaton (CRA), capable of modeling any reward function expressible as a formal language. This mitigates the limitations of existing approaches that they can only handle regular language expressions, whereas this work permits the use of unrestricted grammar. Hence, this framework can handle a larger set of tasks than the existing works. The authors have also demonstrated that this framework supports using LLM-based formal language definitions from natural language task descriptions.

Strengths:
* This paper is well-written and well-explained.
* The CRA automata is novel and can model reward functions expressible as recursively languages.
* This paper also represents a sample efficient learning technique.
* The framework also supports the use of LLM to translate the NL task description into a formal language.

Weaknesses:
* Even if this paper demonstrates a very strong theoretical analysis of the contributions. However, it has lack of empirical studies to showcase the benefits.
* Interested to see how ChatGPT can be used to generate formal language specification from NL? what are the limitations? Very less amount of details have been provided. What kind of quality has been performed over automatic FL generation using ChatGPT?

---

### Decision · Program_Chairs · 2023-12-11

Accept